# Effects of Combined Horizontal Plyometric and Change of Direction Training on Anaerobic Parameters in Youth Soccer Players

**DOI:** 10.3390/sports11020027

**Published:** 2023-01-26

**Authors:** Yiannis Michailidis, Panagiotis Venegas, Thomas Metaxas

**Affiliations:** Laboratory of Evaluation of Human Biological Performance, Department of Physical Education and Sport Science, Aristotle University of Thessaloniki, New Buildings of Laboratories, University Campus of Thermi, 57001 Thessaloniki, Greece

**Keywords:** soccer, horizontal plyometrics, agility, sprint, youth

## Abstract

The aim of this study is to investigate whether the combination of soccer training, plyometric training (PT), and change of direction (COD) exercises would enhance anaerobic performance to a greater extent than training on its own in youth U17 soccer players. Twenty youth players participated in this study. Players were randomly separated into two groups: the control group (CG, *n* = 9) and the intervention group (EX), which performed extra PT and COD exercises (EX, *n* = 11). The duration of the training program was six weeks. Sprint 10 m, 30 m, countermovement jump (CMJ), single leg countermovement jump (CMJ right and left), squat jump (SJ), 505 test, and Illinois agility test were measured pre and post of the training program. The performance in the 505 test improved for the EX group (right leg: *p* = 0.031, left leg: *p* = 0.004). In addition, Illinois test performance increased in the EX group (2.9%, *p* = 0.019). The performances of the two groups differed significantly in the Illinois agility test (*p* = 0.001). This study supports that a short-term combined program of PT and COD exercises can improve change of direction ability in youth U17 soccer players. The lack of effect of the intervention program on sprint and jump performance may be due to the type and volume of plyometric exercises used. The results reflect the training principle of specialization of stimulus. The improvement in performance was presented in tests that had similar characteristics to training stimuli.

## 1. Introduction

Soccer is an intermittent-type sport that incorporates actions with low and high intensity and duration [1]. At developmental ages competing for 11 vs. 11, they can cover 7–8 km [2,3]. Soccer players also perform many intense actions, such as changes of direction, accelerations, decelerations, jumps, and sprints [4,5]. Reilly, Bangsbo, and Franks (2000) [6] have mentioned that explosive actions discriminate between a successful and unsuccessful performance. The above actions depend on the power of the soccer player. Two of the training contents used to improve power factors are plyometric training (PT) and sprints with changes of direction [7,8,9].

PT is used to improve the power of athletes and actions such as jumps and sprints [7]. All these actions are based on the stretch-shortening cycle (SSC). A definition given for the SSC is the ability of the nervous and musculotendinous systems to produce maximum force in as short a time as possible [10]. SSC is the physiological function of the transition from eccentric contraction to concentric [11,12]. The exercises that activate the SSC utilize the elastic properties of connective tissue and muscles, allowing the muscle to store elastic energy during the deceleration phase and release it later during the positive acceleration phase improving performance [12]. In addition, PT is an effective way to improve the rate of strength development that can affect performance in power energies [12].

Plyometric training in adults is well established and improves parameters such as jumping ability speed, acceleration, and explosive force [13]. However, among young people, the use of plyometrics was questioned until a few decades ago due to the high risk of injury. However, these fears in the last decade began to be overcome, and several studies have observed the effectiveness of PT in improving powerful actions such as sprinting and jumping performance in children [7,8,9]. The above studies [7,8,9] refer to positive effects observed in young soccer players after the implementation of programs with a frequency of once or twice a week.

Plyometric training in soccer is expressed through the use of jumping exercises or hopping and is a natural part of the majority of sports movements (i.e., jumping, kicking). Plyometric exercises have various forms depending on the purpose of a training program. Typical plyometric exercises are Countermovement Jump (CMJ), Drop Jump (DJ), and Squat Jump (SJ). In addition, plyometric exercises can be applied at various intensity levels, ranging from low intensity (two-legged bounce) to very high intensity (one-legged jumps on high obstacles). As for the type of plyometrics in a recent review, it seems that unilateral exercises are more efficient than bilateral exercises [14].

The ability to change direction in soccer is particularly important for the performance of soccer players as they should be able to accelerate, slow down, and change direction in a short period of time [6]. In a previous study carried out in the Premier League, it was observed that footballers performed during a match more than 700 changes of direction [15]. From the frequency of occurrence, we can understand the important role it can play in soccer performance. As for jumps, the level of maximum force and the SSC play an important role in the performance of this ability [16].

The recognition of the importance of this ability has had the effect of attracting the interest of researchers. Thus, previous researchers have reported that this ability should be systematically trained during pre-puberty and adolescence, stating that the plasticity of the nervous system in this phase allows the development of motor programs related to the change of direction [17].

The performance in COD of a soccer player depends on various factors such as his strength, power, technique, and anthropometric characteristics [17]. This fact is confirmed by recent studies that implemented force programs or holistic strength programs [9,18] and observed an improvement in COD. Positive effects were also observed from intervention programs that used only COD exercises [19,20]. The use of COD exercises is now very widespread among soccer coaches. However, there is a great variety in the characteristics of these exercises, such as distances, number of changes of direction, and angles of change of direction [9]. Most often, coaches use combined training methods to save time that will be used in the technical-tactical training of soccer players [21]. So, plyometric exercises are very often combined with sprinting and change of direction exercises. In recent years, researchers’ intervention programs combine plyometric exercises with short sprints or with sprints and changes of direction [22,23,24,25], observing positive effects on the performance of young athletes (U17–U19) in anaerobic tests. However, the studies that applied a combined plyometric and sprint program with changes of direction in young soccer are limited, which may be due to the high intensity for which both types of training are characterized. Additionally, the very wide variety of intervention programs and the different effectiveness they may have in different age groups does not allow us to compare them, while at the same time, there is a large gap in the literature on the effectiveness of specific programs in specific age groups. So this is another reason that reinforces the view that these kinds of studies are limited. Therefore, changing some characteristics of intervention programs, such as frequency, duration, type of exercises, and level of players, may alter their effectiveness [26].

The aim of the study was to investigate the effect of a combined program of PT and COD exercises on anaerobic markers of young U17 soccer players. We assumed that applying the intervention program two times a week for 1.5 months would improve the performance of the players on anaerobic indicators (jump performance, sprint performance, acceleration performance, and change of direction performance) compared to the implementation of the usual training program of the control group (CG).

## 2. Materials and Methods

### 2.1. Participants

To determine the smallest number of the study sample, we applied a power analysis based on the design of previous studies [9]. This analysis showed that in order to show the interaction between the two groups and the time points, at least 14 subjects had to participate (effect size of >0.55, probability error of 0.05, power of 0.95). Initially, thirty-five youth soccer from a local academy were approached to participate in the study. Of these, thirty accepted, but only twenty of them met the participation criteria, which were (1) the lack of musculoskeletal injuries in the last year, (2) to have participated in more than 95% of the total workouts, (3) not to be experienced in plyometric exercises, (4) not to take medications. The players trained four times a week for 1.5 h and participated in one match every weekend. The participants came from all competitive positions, with the exception of the goalkeeper position. Goalkeepers were excluded from the study.

Participants and their parents were informed of the benefits and potential risks of the study, and the parents signed a consent form. The local Institutional Review Board approved the study in the spirit of the Helsinki Declaration. Participants’ characteristics are presented in Table 1.

### 2.2. Design

The independent variables of the study were soccer training and the combined program of PT and COD drills. Anthropometric and fitness assessment (countermovement jump [CMJ], single countermovement jump with the right leg [CMJR], single countermovement jump with the left leg [CMJL], squat jump [SJ], acceleration and speed [10 m and 30 m], change of direction [505 test and Illinois agility test], were the depended variables. In the study, a random, 2 groups, repeated measures experimental design was used. More specifically, the players were initially divided according to their competitive position, and then in a random way, the two groups were formed, including players from all competitive positions. In this way, the control (CG, *n* = 9) and intervention (EX, *n* = 11) groups were formed.

The study was carried out during the in-season period, and the intervention lasted six weeks. The CG carried out only the conventional soccer practice, while in the EX, a part of the soccer training was replaced by the intervention program. The training duration of the two groups was the same. Two weeks before the start of the intervention, the EX group was familiarized with the plyometric exercises and the change of direction exercises. Additionally, all the players were informed about the tests in order to minimize the learning effect error. The body mass, height, and body fat percentage of the players were measured during their first visit. The rest of the tests were carried out in the next two visits. The measurements were repeated after six weeks of intervention. All measurements were made in the same order. At the beginning of each training session, a 15-min warm-up took place, and at the end of the training, a 10-min cool-down period was performed. During the study, all participants participated in six matches. In addition, the participants consumed water ad libitum to ensure proper hydration during training and testing.

### 2.3. Methodology

#### Plyometric Training Protocol

The duration of the intervention program was six weeks. As mentioned above, at the beginning of each training session, warm-ups were carried out. Each group participated in four training sessions per week and one match. The training included technique and tactical exercises as well as small-sided games. The intervention program was implemented immediately after the warm-up to ensure full neuromuscular activation and lasted 15–20 min. The intervention program was applied twice a week, with an intermediate rest time of 72 h. The total duration of the workouts was 90 min. The three exercises that were used are presented in Figure 1A. In the first three weeks, the number of jumps and measures in the changes of direction increased. In the fourth week, the volume was reduced to make it easier to make adjustments, and over the next two weeks, the progressive increase in volume continued. The final measurement was made five days after the last training with plyometrics and COD. The progressive change in the volume of the intervention program was based on previous studies [27]. The study was carried out on a field with synthetic grass. More details about the number of jumps and the distances of the COD program are presented in Table 2.

### 2.4. Anthropometric and Assessment of Maturity Status

Body weight and height were measured with an accuracy of 0.1 kg and 0.1 cm, with the participants wearing their underwear and without shoes (Seca 220e, Hamburg, Germany). The percentage of body fat was calculated by measuring four skinfolds (biceps, triceps, suprailiac, subscapular) on the right side of the body, and the percentage of body fat was estimated with the equation proposed by Siri [28].

### 2.5. Speed Testing

To measure the speed, distances of 10 m and 30 m were used. The participants, from an upright starting position and 0.3 m behind the starting line, started whenever they wanted to cover the distance of 30 m as fast as possible, passing through three gates of photocells (Microgate, Bolzano, Italy) [9]. The photocells were placed at the height of 0.6 m to avoid incorrect signals from the movement of the hands [29]. The coefficient of variation for test-retest trials was 3.2%.

### 2.6. Vertical Jump Testing

The participants performed 3 kinds of jump tests: (a) SJ: participants from a stationary semi-squatted position (90° angle at the knees) performed a maximal VJ; (b) CMJ: participants from an upright standing position performed a fast-preliminary motion downwards by flexing their knees and hips followed by an explosive upward motion by extending their knees and hips; (c) single leg CMJ: participants from an upright standing position to one leg, performed a fast-preliminary motion downwards by flexing their knees and hips followed by an explosive upward motion by extending their knees and hips [30]. All tests were performed with the arms akimbo. The jump height was measured using the electronic leap mat Chronojump of Boscosystem (Chronojump, Boscosystem, Barcelona, Spain). The coefficients of variation for test-retest trials were 3.3%, 3.1%, 3.6%, and 3.8% for SJ, CMJ, CMJR, and CMJL, respectively.

### 2.7. Illinois Agility Test

Participants start from an upright position. They sprint from A to B and from there to C. They zig-zag up to D and turn in the same way to C. Then they sprint to E and from there to F, where they complete the route. Photocells are positioned in positions A and F (Microgate, Bolzano, Italy) (Figure 2) [31].

### 2.8. 505 Test

Participants sprint from point A for 15 m to point C, where they turn 180° and sprint up to point B (Figure 2B) [7]. This route is performed twice by using a turn of 180° the right foot and twice by using a turn of 180° the left foot (used the best time). There is a photocell (Microgate, Bolzano, Italy) in B that records the time the participant takes from B to C and back to B. The coefficients of variation for test-retest trials were 3.4% for the right leg and 3.6% for the left leg.

### 2.9. Statistical Analysis

Data are presented as means ± SD. Furthermore, for fitness variables, the confidence intervals (CI) were mentioned. Data normality was verified with the 1-sample Kolmogorov-Smirnoff test; therefore, a nonparametric test was not necessary. Data were analyzed by a 2-way repeated measures (trial × time) analysis of variance with planned contrasts on different time points based on the 2-group, repeated measures experimental design that was used in the study. When a significant effect was found, Bonferonni post hoc correction was performed. The level of significance was set at *p* < 0.05. As mentioned earlier, a power analysis was performed to estimate the smallest acceptable number of participants to analyze the interaction between the group and time points of measurements. The SPSS version 25.0 was used for all analyses (SPSS Inc., Chicago, IL, USA).

## 3. Results

The results of the statistical analysis showed that there were no differences in any of the independent parameters between the two groups at the start of the study. In addition, at the end of the study, there were no changes in the anthropometric characteristics of the participants in either group (Table 1).

In the jumps, there were no differences between the pre and post-measurements in the two groups. In addition, the two groups did not differ from each other in post-measurements. The statistical indicators are presented in Table 3, while the test performance is presented in Figure 3.

No changes were observed in the intervention group nor the control group between the measurements before and after for 10 m and 30 m tests. The two groups did not differ from each other in the speed tests in the measurement after the intervention. The statistical indicators are presented in Table 3, while the test performance is presented in Figure 4.

EX group improved its performance in the Illinois agility test (2.9%). No differences were observed for CG between the two measurements (−0.8%). Differences between groups were observed at the post-measurement. In addition, the EX group improved its performance in the 505 test for both legs (Improvement: Right: 3.75%, Left: 2.9%). The statistical indicators are presented in Table 3, while the test performance is presented in Figure 5.

## 4. Discussion

The purpose of this study was to investigate the effect of a short plyometrics program combined with COD drills on anaerobic indicators on U17 young soccer players during the season. After six weeks of intervention, the EX group improved in the change of direction tests. As mentioned above, differences between the groups were observed only in the Illinois agility test. No other significant differences were observed.

Previous studies report that plyometric exercises improve jumping ability in young soccer players [9,32]. Plyometric exercises are known to cause adjustments to nerve factors such as the level of neuromuscular activation, motor intermuscular and intramuscular coordination, and elastic advantages of SSC [33]. In the present study, no changes in jumping ability were observed in any of the considered jumps (CMJ, single leg CMJ, SJ). This is probably due to a limited affinity between the type of jumps used for training (horizontal jumps) and the jumps used to measure the effect (vertical jumps).

In contrast to the present study, Aloui, Souhail, Hayes, Bouhafs, Chelly, and Schwesig (2021) [22] observed an improvement in jumping horizontal and vertical ability in U18 soccer players. Similar adaptations are mentioned by Hammami, Gaamouri, and Suzuki (2020) [24] and Sáez de Villarreal, Suarez-Arrones, Requena, Haff, and Ferrete (2015) [34]. Here it should be mentioned that in the aforementioned studies, during intervention training, vertical jumping exercises were used as well as tests for the assessment of horizontal jumping ability.

In soccer, there are many jumping movements and many sprints. In all these actions, an important factor is a stretch-shortening cycle (SSC) [35]. This physiological factor is trained with plyometric exercises and can be distinguished into fast SSC (<250 ms) and slow (>250 ms) SSC. Plyometric exercises are used to improve power (i.e., to exert maximal force in as short a time as possible) [36] and show a wide variety. However, coaches try to include exercises that have more in common with the sport. Thus, it has been reported that adaptations to vertically (VPT) and horizontally oriented PT (HPT) will transfer better to athletic tasks that are carried out in the same direction as they are performed [37].

In addition, Dello, Martone, Milic, and Padulo (2017) [38] observed the effect of training specificity on the final results, concluding that horizontal jumps improved sprints and changes of direction, while vertical jumps improved CMJ performance. In the present study, this observation of the researchers seemed to be verified.

Sprinting is one of the most essential actions of a soccer player that affects the outcome of a match. For this reason, it is a factor that is studied extensively. Recent literature reports the positive effect of plyometrics on speed indicators [8,9,39] and combined programs of plyometrics with a change of direction sprints [22,23,25,39]. However, in the present study, no improvement was observed at 10 m and 30 m.

A non-significant effect of plyometric exercises on sprints and their combination with COD is also reported by other studies for 30 m [9,40]. These differences between studies are likely due to the different levels of participants, the different training exercises used, and the different characteristics of the exercise and assessment protocols used by each study. As far as the present study is concerned, the lack of significant effects of plyometrics may be due to insufficient stimulus.

In the present study, significant effects of the intervention program were observed in the two direction change tests (505 test and Illinois agility test). In addition to the improvement observed in the Ex group, a difference also appeared between the two groups (EX and CG in the Illinois agility test). The COD tests used included accelerations and decelerations. Performance in these tests, therefore, depends on factors that affect the SSC. It has been reported that muscle predisposition during the eccentric phase of the SSC stores elastic energy, which can improve performance during the concentric phase that follows using this extra energy [41]. Thus, by lengthening the muscles during the deceleration phase, before changing direction, energy can be stored that is attributed to the next phase during acceleration. Therefore, the intervention program may have improved the function of the SSC. However, the function of the SSC, according to a recent study, may be influenced by other factors such as pre-activation, cross-bridge kinetics, and residual force enhancement [42]. The above are hypotheses unless in vivo investigations are made into the physiological mechanisms on which the function of the SSC depends are unknown, and we cannot confirm or reject them. However, this was not the aim of this study. The findings of the present study agree with those of recent research [9,22,23,25]. However, there are also studies that did not observe an improvement in the performance of COD ability after a plyometric program combined with COD exercises [7].

Generally, the ability to change direction can be associated with neural adaptations and improved motor unit recruitments [43], with an improvement in the rate of force development [44] and movement efficiency [45]. All the above adaptations may be the result of the plyometric exercises but also the specialized exercises with changes of direction used during the six weeks of intervention. In addition, the relevance of the training stimuli (e.g., in the third exercise, the direction changes were similar to those of the 505 test) with the measurement tests may have led to these results.

## 5. Limitations

The present study involved only male youth soccer players under the age of 17, so the results cannot be generalized to other developmental ages and gender. Added to this is the fact that the sample was limited. A limited number of field tests and a specific intervention protocol were also used. Therefore, future research in both sexes, at different developmental ages, in as large samples as possible, more field tests and laboratory tests, and applying different intervention protocols will give a clearer and more comprehensive picture of the effect of combined plyometric programs with COD on anaerobic indices of young soccer players.

### Practical Applications

The results of this study showed that a combined program of PT and COD exercises could improve performance in changing direction of young U17 soccer players. The intervention program was short and can be introduced into the training sessions with the aim of improving the ability to change direction.

## 6. Conclusions

In conclusion, this study supports that a short-term combined program of PT and COD drills can offer some improvements in the change of direction ability of youth U17 soccer players. The absence of training effect for sprints and jumps could be the result of the level of the participants, the duration of the program, and the kind and volume of plyometric exercises that were used.

## Figures and Tables

**Figure 1 sports-11-00027-f001:**
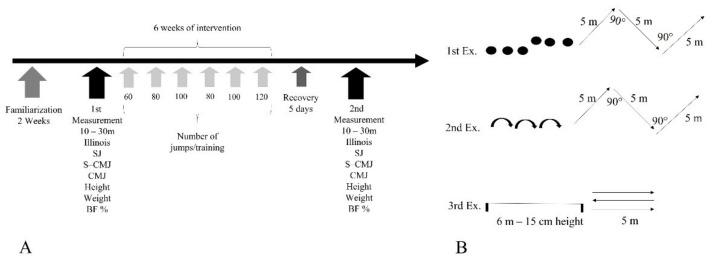
(**A**) Design of the study. (**B**) Description of the three exercises.

**Figure 2 sports-11-00027-f002:**
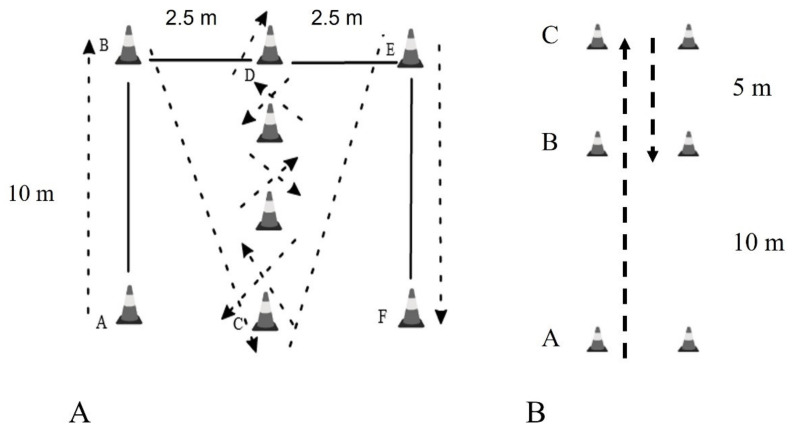
Description of (**A**) Illinois agility test and (**B**) 505 test.

**Figure 3 sports-11-00027-f003:**
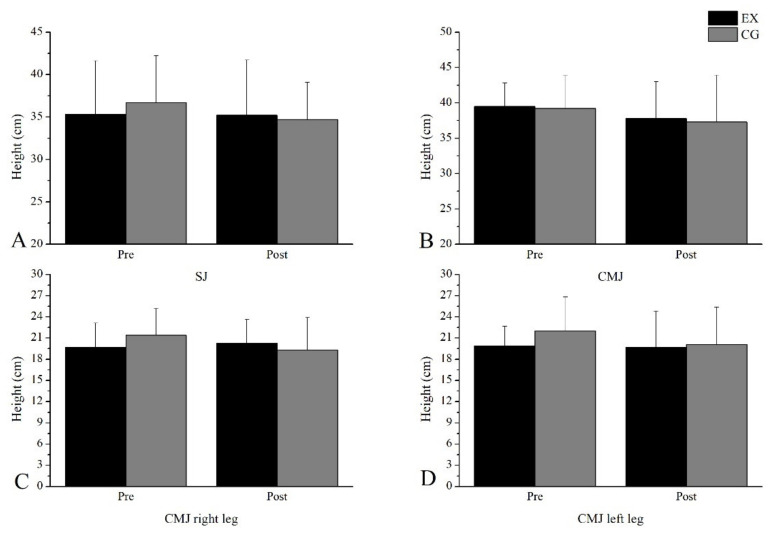
Performance changes in (**A**) SJ, (**B**) CMJ, (**C**) CMJ right leg, (**D**) CMJ left leg.

**Figure 4 sports-11-00027-f004:**
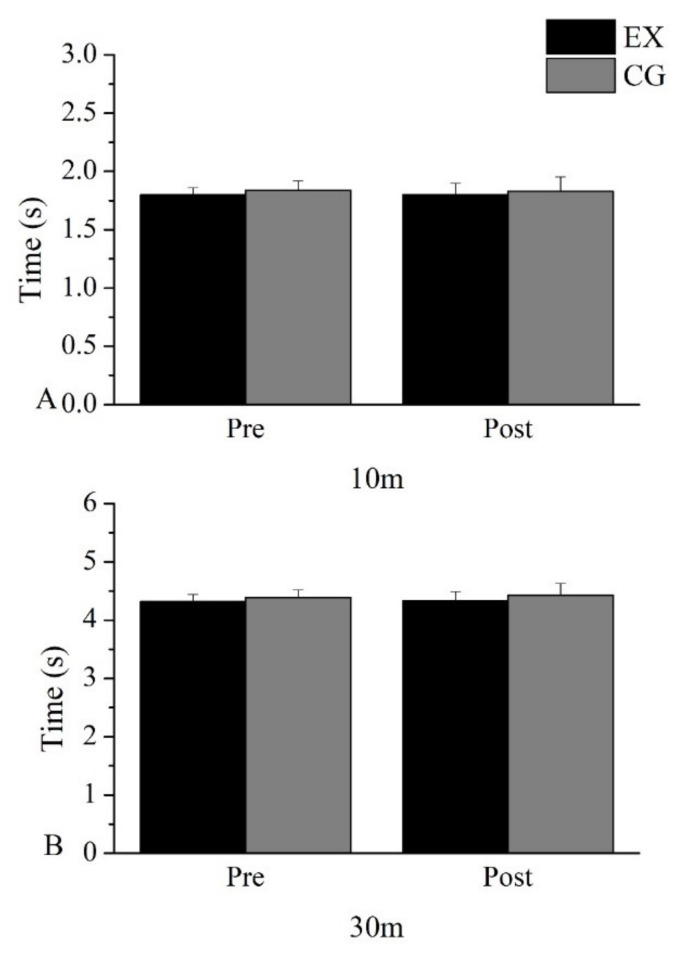
Performance changes in (**A**) 10 m, (**B**) 30 m.

**Figure 5 sports-11-00027-f005:**
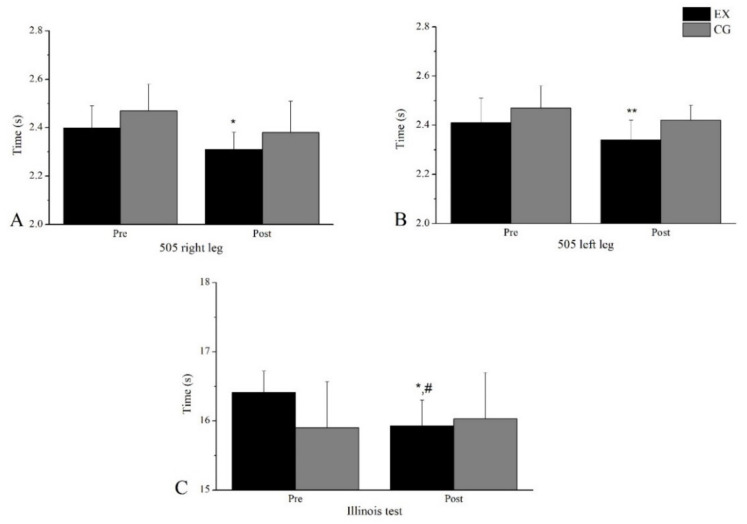
Performance changes in (**A**) 505 right leg, (**B**) 505 left leg, (**C**) Illinois agility test. * denotes a significant difference with Pre (*p* < 0.05). ** denotes a significant difference with Pre (*p* < 0.01). ^#^ denotes a significant difference between groups (*p* < 0.05).

**Table 1 sports-11-00027-t001:** Participants’ physical characteristics.

	CG (*n* = 9)		EX (*n* = 11)	
	Pre-Training	Post-Training	Pre-Training	Post-Training
Age (years)	16.7 ± 0.1	16.8 ± 0.1	16.5 ± 0.3	16.6 ± 0.3
Training age (years)	10.3 ± 1.4	10.4 ± 1.4	9.8 ± 0.6	9.9 ± 0.6
Height (cm)	176 ± 8	176 ± 9	176 ± 4	176 ± 4
Weight (kg)	72.6 ± 9.6	72.6 ± 10.4	69.9 ± 7.2	70.3 ± 7.1
Body fat (%)	15.3 ± 2.7	15.2 ± 2.7	13.8 ± 2.2	12.3 ± 2.1

**Table 2 sports-11-00027-t002:** Program characteristics.

Week	Exercise 1	Exercise 2	Exercise 3	No. of Jumps	Meters of COD
1st	3 set × 3 reps	3 set × 3 reps	2 set × 6 reps	60	120
2nd	4 set × 4 reps	3 set × 4 reps	2 set × 6 reps	80	135
3rd	4 set × 5 reps	4 set × 4 reps	2 set × 7 reps	100	150
4th	4 set × 4 reps	3 set × 4 reps	2 set × 6 reps	80	135
5th	4 set × 5 reps	4 set × 4 reps	2 set × 7 reps	100	150
6th	5 set ×5 reps	4 set × 5 reps	3 set × 5 reps	120	180

Reps: repetitions.

**Table 3 sports-11-00027-t003:** Statistical results of measurements.

Test		Trial	Interaction	CI: Time		CI: Trial	
				Pre	Post	EX	CG
10 m (s)	F = 0.116	F = 1.008	F = 0.001	1.783–1.849	1.771–1.848	1.739–1.838	1.764–1.911
	*p* = 0.742	*p* = 0.345	*p* = 0.972				
	η^2^ = 0.014	η^2^ = 0.112	η^2^ = 0				
30 m (s)	F = 0.609	F = 0.914	F = 0.296	4.299–4.418	4.314–4.446	4.227–4.426	4.279–4.547
	*p* = 0.458	*p* = 0.367	*p* = 0.601				
	η^2^ = 0.071	η^2^ = 0.103	η^2^ = 0.036				
Illinois test (s)	F = 8.499	F = 4.183	F = 22.699	15.874–16.439	15.675–16.236	15.883–16.410	15.435–16.498
	*p* = 0.019	*p* = 0.036	*p* = 0.001				
	η^2^ = 0.515	η^2^ = 0.050	η^2^ = 0.739				
505 right leg (s)	F = 6.818	F = 2.820	F = 0.272	2.379–2.466	2.298–2.396	2.289–2.393	2.347–2.509
	*p* = 0.031	*p* = 0.132	*p* = 0.616				
	η^2^ = 0.460	η^2^ = 0.261	η^2^ = 0.033				
505 left leg (s)	F = 16.362	F = 2.911	F = 0.906	2.431–2.469	2.331–2.410	2.312–2.434	2.400–2.494
	*p* = 0.004	*p* = 0.126	*p* = 0.369				
	η^2^ = 0.672	η^2^ = 0.267	η^2^ = 0.102				
SJ (cm)	F = 0.460	F = 1.781	F = 0.283	32.021–38.371	31.588–36.604	30.640–36.610	33.081–38.253
	*p* = 0.517	*p* = 0.219	*p* = 0.609				
	η^2^ = 0.054	η^2^ = 0.182	η^2^ = 0.034				
CMJ (cm)	F = 1.596	F = 0.003	F = 0.034	37.609–41.075	33.326–41.099	36.274–40.388	34.316–42.131
	*p* = 0.242	*p* = 0.955	*p* = 0.858				
	η^2^ = 0.166	η^2^ = 0	η^2^ = 0.004				
CMJ right leg (cm)	F = 1.090	F = 0.154	F = 1.628	18.402–22.547	17.533–21.675	18.082–21.371	17.066–23.638
	*p* = 0.327	*p* = 0.705	*p* = 0.238				
	η^2^ = 0.120	η^2^ = 0.019	η^2^ = 0.169				
CMJ left leg (cm)	F = 2.548	F = 0.262	F = 0.831	19.423–22.670	17.494–22.158	17.155–22.437	17.056–25.097
	*p* = 0.149	*p* = 0.622	*p* = 0.389				
	η^2^ = 0.242	η^2^ = 0.032	η^2^ = 0.094				

## Data Availability

Data are available upon request from the corresponding author.

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
