# Peer review of "Effects of Combined Horizontal Plyometric and Change of Direction Training on Anaerobic Parameters in Youth Soccer Players"

_sports, 2023, doi:10.3390/sports11020027_

Round 1
Reviewer 1 Report
Dear Editor,
Thank you for the opportunity to evaluate the manuscript “Effects of combined horizontal plyometric and change of direction training on anaerobic parameters in youth soccer players”. The objective of the research was: to investigate whether the combination of soccer training, plyometric training (PT), and change of direction (COD) exercises would enhance anaerobic performance to a greater extent than training on its own in youth U17 soccer players. The sample was composed of twenty youth players. After reading the manuscript, I suggest the following changes:
1 – In the introduction, there is little about the subject (PT and COD);
2 – The authors state that research on PT and COD is limited. However, they do not present the limitations or findings of research in this area;
3 – The hypothesis (We assumed that applying for the intervention program 2 times a week for 1.5 months would improve the performance of the players on anaerobic indicators (jump performance, sprint performance, acceleration performance, and change of direction performance) compared to the implementation of the usual training program of the control group.) has no theoretical support.
4 – The authors need to better characterize the sample. How many hours a week did the athletes train? How many games did the athletes play each week? What are the players' positions?
5 – How were the players chosen for each group? Did the authors consider the positions of each player to make the distribution in the groups? It is necessary that the groups are balanced by position, since the training respect the specificity of the position in the game.
6 – Knowing that the group played 6 games, it is necessary to know: Which athletes played more and which athletes played less time? These athletes were part of which group analyzed? When considering training, it is known that first-team athletes train longer than substitute athletes. In addition, starting players are placed in more challenging and motivating situations than substitutes. How did the authors balance the groups in relation to the starting athlete and the substitute athlete?
7 – For each of the tests performed, reference authors must be indicated.
8 – In the discussion, the authors show that there was a mistake between the training performed and the selected test. This shows that there was no coherence between the training and the analyzed variables. This fact has already been demonstrated in the literature, as the authors themselves mentioned.
9 – The authors did not discuss the COD results in depth. This needs to be redone.
10 – The practical applications are obvious. I suggest that authors, as far as possible, make suggestions beyond the obvious.
11 – Only 11 references are from 2018 or more recent. On the other hand, 31 references are older than 2018. I suggest authors seek more recent theoretical references.
Given the above, I am in favor of major corrections.
Author Response
Dear Reviewer,
We are grateful for your comments to our manuscript. We have addressed all the comments as shown in the revised manuscript.

Reviewer 2 Report
General Comments
First, I would like to recognize the authors for the data they collected to assess the effectiveness of their six-week intervention in youth U17 soccer players. Despite the small sample size, I still find the results significant to be published. Furthermore, I find the field testing enough and accurate for the purpose of this study. I would love to see the intervention implemented in younger age groups as those players do not perform poorly on the agility tests (especially when the agility test is performed with the ball).
Specific comments
Abstract: The abstract contains all the information that is needed.
Introduction (try to have the same font throughout the whole manuscript)
‘Two of the training contents used to improve power factors are plyometric training (PT) and sprints with changes of direction’. You need to add scientific support for this statement. Studies 10-12 that were reported later can be included here as well to support the statement.
‘PT used to improve the power of athletes and actions such as jumps and sprints’. I would say is used to…. Also, here you need to provide scientific evidence (previously published studies that support that statement).
‘A definition given for the SSS is the ability of the nervous and musculotendinous system’. I guess you mean SSC (stretch-shortening cycle)?
Materials and methods
‘This analysis showed that in order to show the interaction between the two groups and the time points, at least 14 subjects had to participate (effect size of >0.55, probability error of 0.05, power of 0.95)’. At least 14 subjects in EACH group.
‘The total duration of the workouts was 90 minutesThe three’. Separate the sentences
Results
In Table 3 include the units of measurement for each parameter.
Discussion (use the same font here as well).
The discussion is very well written and supported.
Limitations are accurately presented.
Author Response

(The authors gave the same response as above.)

Round 2
Reviewer 1 Report
Accept in present form